# Total Sealing Technique: A Preliminary Study on a Novel Surgical Approach That Significantly Reduces the Incidence of Upper Extremity Lymphedema Following Axillary Dissection in Patients with Breast Cancer

**DOI:** 10.3390/cancers17081285

**Published:** 2025-04-10

**Authors:** Naoya Ikeda, Takuya Nagata, Teiji Umemura, Manabu Watanabe

**Affiliations:** 1Department of Surgery, KIWA Hospital, Wakayama 648-0085, Japan; umemura.teiji@nanroukai.or.jp; 2Department of Forensic Medicine, Wakayama Medical University, Wakayama 641-8509, Japan; 3Department of Surgery, Toho University Ohashi Medical Center, Tokyo 153-8515, Japan; takuya.nagata@med.toho-u.ac.jp (T.N.); manabu@oha.toho-u.ac.jp (M.W.)

**Keywords:** breast cancer-related lymphedema, axillary lymph node dissection, total sealing technique, bipolar-vessel-sealing system, LigaSure™ Exact Dissector

## Abstract

Breast cancer-related lymphedema (BCRL), characterized by lymphatic fluid accumulation in the upper extremities following breast cancer treatment, significantly affects the quality of life of survivors of breast cancer. In a previous study, we introduced a novel surgical procedure, the Total Sealing Technique (TST), which utilizes a bipolar-vessel-sealing system (LigaSure™ Exact Dissector, Medtronic plc, Dublin, Ireland) to reduce lymphatic leakage during axillary lymph node dissection (ALND) for breast cancer. After observing that patients who underwent this procedure had fewer cases of BCRL, we evaluated its incidence. A comparative analysis of 36 patients who underwent conventional electrocautery (CONV) and 35 who underwent the TST revealed that the TST significantly reduced the incidence of BCRL. Once the surgeon becomes accustomed to it, the TST becomes a simple surgical procedure that provides reliable outcomes regardless of the surgeon, timing, or location, suggesting that it has the potential to transform axillary dissection in breast cancer surgery.

## 1. Introduction

The primary goal in treating patients with breast cancer who have positive axillary lymph nodes is to achieve optimal tumor control while minimizing side effects that include breast cancer-related lymphedema (BCRL). However, despite notable efforts to reduce axillary surgery, BCRL remains a common complication of locoregional treatment [1], significantly affecting both individuals and the healthcare system. In addition, lymphedema is difficult to cure, and its management is costly, therefore placing an unfair financial burden on patients. In 2020, Naoum et al. conducted a prospective study of 1815 patients and reported that the risk of BCRL correlated with the extent of axillary surgery. The authors classified patients according to the extent of axillary surgery with or without regional lymph node radiation (RLNR), SLNB alone, SLNB plus RLNR, axillary lymph node dissection (ALND) alone, and ALND plus RLNR, and found a five-year cumulative BCRL incidences of 8.0%, 10.7%, 24.9%, and 30.1%, respectively [2]. Notably, recent data show that more than one in four patients who undergo axillary dissection develop BCRL. In response to this challenge, various nonsurgical and surgical strategies have been developed to prevent and treat BCRL effectively [3]. Axillary Reverse Mapping (ARM), a surgical approach used to prevent lymphedema, was introduced by Klimberg in 2008 and is an improved lymph node mapping technique that identifies the SLN while preserving upper extremity function to minimize the risk of lymphedema [4,5]. ARM was designed to reduce the risk of lymphedema by identifying and preserving lymphatic drainage pathways of the arm in patients undergoing ALND or sentinel lymph node biopsy (SLNB) [6]. Immediate lymphatic reconstruction (ILR) is a surgical technique designed to restore the lymphatic drainage of surgical extremities. ILR has been increasingly discussed in the literature; however, it requires further investigations for more supporting evidence [7]. In practice, performing these complex procedures in patients with breast cancer requiring axillary dissection is quite challenging. Most importantly, there is an urgent need for the development of a simple surgical procedure that provides consistent results regardless of when, where, or who performs the procedure to reduce the risk of BCRL. This study builds on our previous research [8] that compared surgical outcomes between the Total Sealing Technique (TST) and conventional electrocautery (CONV) groups. The previous study demonstrated that the use of a bipolar-vessel-sealing system (LigaSure™ Exact Dissector (LGSED)) reduced lymphorrhea and seroma formation. Building on these findings, the current study aims to evaluate whether the TST using a vessel-sealing device (LGSED) improves the incidence of breast BCRL compared to conventional axillary dissection using a monopolar electrocautery scalpel.

## 2. Materials and Methods

### 2.1. Patients

A previous study was conducted on a consecutive series of 71 patients who underwent total mastectomy and ALND at Nara Medical University between 1 December 2015 and 30 November 2021. The patients were divided into two groups according to the surgical procedure: the conventional electrocautery (CONV) and the TST group. Several factors were compared between both groups, including intraoperative blood loss, operative time, total drainage volume, mean number of days to drain removal, postoperative hospital stay, mean time from surgery to induction of postoperative chemotherapy, and postoperative complications [8]. During the follow-up period, most patients who underwent TST did not develop BCRL. However, there was no comparison of the level of incidence of BCRL between both groups in the previous study. Therefore, in this current study, we performed a comparative analysis of the incidence of BCRL between these two groups. Additionally, we investigated the relationship between BCRL and various factors that may influence its incidence, including age, body mass index (BMI), postoperative radiotherapy, neoadjuvant and/or postoperative chemotherapy, use of a taxane-containing regimen, number of dissected lymph nodes, total drainage volume, duration of drainage, and incidence of seroma in each group. In both groups, age, BMI, number of dissected lymph nodes, total drainage volume, and drainage duration were stratified based on median values. Subsequently, all patients were followed up with until 31 December 2024, with a median follow-up period of 74.6 months (range: 39.1–110.3 months). Specifically, the median follow-up period was 95.6 months (range: 74.4–110.3 months) and 61.6 months (range: 39.1–70.1 months) in the CONV and TST groups, respectively.

### 2.2. Surgical Technique and Axillary Drain Management

Total mastectomy was performed with an electrocautery scalpel, creating skin flaps extending 2–3 cm below the subclavian bone and beyond the inferior breast margin. Axillary lymph node dissection (ALND) was performed through the same incision. In the TST group, all dissected tissue was sealed with LGSED prior to resection, eliminating the need for instrument exchange, sutures, or electrocautery. In contrast, the CONV group used electrocautery or suture ligation for hemostasis. In the TST group, the breast margin near the axilla was resected after sealing with LGSED to prevent lymphatic leakage using the same technique as for ALND. In ALND, level I and II lymph nodes were dissected, and the remaining side was double sealed and resected in the TST group. A closed suction drain was placed in the axilla and anterior chest, and the wounds were sutured with absorbable monofilament sutures. The axillary drain was removed when daily output was <50 mL, and patients were discharged the day after drain removal unless drainage was uncontrolled. Nurses monitored drainage daily and recorded complications, and patients attended follow-up visits every one to two weeks for at least 30 days.

### 2.3. Lymphedema Education

All patients who underwent ALND were educated on lymphedema following discharge by a certified breast care nurse in an outpatient setting. This education included the causes, symptoms, diagnoses, treatment options, and prevention of lymphedema. Prevention strategies emphasize avoiding heavy lifting and infection, exercising regularly, and using compression garments.

### 2.4. Breast Cancer-Related Lymphedema Sreening and Definition

BCRL measurements were performed according to the Japanese Lymphedema Society Guidelines [9,10]. The following preoperative measurements were taken from both upper extremities to establish baseline values and assess left-right differences: 1. Palmar diameter along the line connecting the metacarpophalangeal joints of the second through fifth fingers; 2. circumference of the wrist; 3. the circumference 5 cm distal to the cubital fossa; and 4. the circumference 10 cm proximal to the cubital fossa. To assess the incidence of BCRL, all patients who underwent axillary dissection were subsequently assessed at the above-mentioned upper extremity sites during their outpatient visits every 3 to 12 months. Assessment of BCRL was conducted as follows: pre-treatment circumference was measured, and post-treatment comparative observations were made on the same side and area. BCRL was diagnosed when a difference of 2 cm or more was observed between the affected and non-affected side [11].

### 2.5. Statistical Analysis

Categorical variables were compared using the chi-square test or Fisher’s exact test. Fisher’s exact test was used when the observed data contained zero values to ensure statistical validity, as the expected frequencies in some cells were less than five. Continuous variables were analyzed using the Student’s *t*-test. Statistical significance was set at *p* < 0.05, and the JMP16 software package (SAS, Tokyo, Japan) was used to perform all statistical analyses.

## 3. Results

The patients’ baseline characteristics are presented in Table 1. The following factors were compared between the CONV and TST groups: age, BMI, postoperative radiation therapy, neoadjuvant chemotherapy, postoperative chemotherapy, taxane-containing regimen, the total number of removed lymph nodes, and the stage. The TST group included significantly more patients with advanced disease stages than in the CONV group; however, no significant differences were observed between both groups regarding other factors.

### 3.1. Incidence of BCRL

As shown in Table 2, the incidence of BCRL was significantly lower in the TST group than in the CONV group (2.9% vs. 22.2%; *p* = 0.028).

### 3.2. Relationship Between BCRL and Various Factors

In the TST group, there was no significant relationship between the occurrence of BCRL and factors including age, BMI, postoperative radiotherapy, neoadjuvant and/or postoperative chemotherapy, taxane-containing regimen, the total number of removed lymph nodes, the total drainage volume, the duration of drainage, or the incidence of seroma (Table 3).

Conversely, a significant relationship was observed between BCRL incidence and age, neoadjuvant and/or postoperative chemotherapy, taxane regimen, and the total drainage volume for the CONV group (Table 4).

## 4. Discussion

Fortunately, ALND procedures are becoming increasingly rare thanks to recent studies in the literature. Recent studies have demonstrated a decreasing need for ALND due to advancements in sentinel lymph node biopsy and neoadjuvant therapies, which help reduce surgical morbidity in breast cancer patients [12,13]. Current data suggest that the onset of BCRL is influenced by multiple factors that can be categorized into three groups: disease- and treatment-related (including tumor size, ALND surgery, chemotherapy, and radiation therapy), lifestyle (such as physical activity, BMI, and preventive behaviors), and demographic factors (monthly income, marital status, and ethnicity) [1,14,15,16,17,18,19]. Data from this study showed that age, neoadjuvant and/or postoperative chemotherapy, and taxane-containing regimens were risk factors for BCRL occurrence. Notably, cases with a total drainage volume of >600 mL had a significantly higher incidence of BCRL, recognized as a newly identified factor. A PubMed search yielded one report on the relationship between the amount of fluid drained from the axillary lymph nodes after axillary dissection and lymphedema. In 2017, Kretschmer et al. reported that a high volume of postoperative drainage fluid after complete lymph node dissection for axillary lymph node metastasis of melanoma significantly correlated with lymphedema [20]. However, the underlying mechanism has not yet been elucidated. Some reports indicate that patients with longer drainage periods have a higher risk of developing BCRL [21]. Similarly, Saadet et al. reported that a prolonged duration of axillary drainage is a risk factor for BCRL [22]. There are two possible explanations for this phenomenon. First, a longer drainage duration reflects a higher degree of lymphatic vessel damage, supporting the notion that extensive axillary surgery increases the incidence of BCRL [23]. Second, patients with drainage tubes are at an increased risk of developing BCRL owing to the restricted movement of the affected limb [24]. A cross-sectional study involving 775 patients supports this hypothesis, showing that women who exercise the affected upper extremity have a reduced risk of developing BCRL, potentially due to the “muscle pump” mechanism [19,25]. A high volume of drainage fluid following ALND for breast cancer may be a predictive factor for BCRL development. The potential effectiveness of compression garments in the prevention and treatment of BCRL has been suggested [26]. However, it is still in the research stage.

Several methods have been reported to measure BCRL, including bioimpedance spectroscopy (BIS), water displacement, circumferential measurement, and infrared techniques. While BIS has been recommended for stage 0 lower extremity lymphedema, its evidence remains limited [26,27]. Infrared techniques have high intra-examiner reliability (ICC intra = 0.99, 95% CI = 0.97–1.00) but lack inter-examiner reliability data, and their cost limits routine use [28,29]. Water displacement is highly reliable for upper extremities but impractical for daily use (ICC intra = 0.99, 95% CI = 0.99, 0.99/SEM = 0.7%) [30]. The most practical method for clinical use is circumferential measurement, which offers comparable reliability (ICC intra = 0.99, 95% CI = 0.98, 0.98) and high validity (SEM = 2.8%) [30]. A difference of ≥2 cm between arms is clinically significant, while a difference of 1 cm may indicate BCRL [31,32].

The treatment of BCRL in practice after development is challenging [27]. Consequently, many nonsurgical and surgical strategies have been developed for BCRL prevention and treatment. Recently, there has been an increased focus on prophylactic surgical procedures performed during initial axillary surgery to prevent BCRL. Since 2007, ARM has been developed as a novel surgical approach to distinguish the lymphatic drainage pattern of the upper extremity from that of the breast [6,28]. This procedure aimed to minimize the risk of lymphedema by identifying and preserving the lymphatic drainage pathways of the arm in patients undergoing ALND or SLNB [6]. By injecting a blue or fluorescent dye into the arm, the technetium-labeled lymphatics of the breast can be visually distinguished from those of the arm, allowing for their preservation during dissection. A large prospective study of ARM reported that the incidence of lymphedema at 26 months was 0.8% and 6.5% in patients who underwent SLNB and ALND, respectively [29]. In our present study, we found that modifying the surgical procedure by performing ALND with the TST resulted in a 2.9% reduction in BCRL incidence, suggesting that this approach provides better results than ALND combined with ARM.

Furthermore, we discuss how the TST using an LGSED improves BCRL incidence compared to conventional axillary dissection performed with a monopolar electrocautery scalpel. The primary reason for this difference likely lies in the extent of thermal diffusion into the surrounding tissues. One study described the extent of thermal spread using LigaSure. In the study, laparotomy was performed on eight pigs, and target vessels/organs were sealed with the LigaSure™ (LS1100) system prior to excision. Thermographic data showed that thermal damage was limited to 1.8 mm from the device, with jaw surface temperatures remaining within the range of surgical use (approximately 35 °C). Moreover, histological studies confirmed minimal thermal damage [30]. Sutton et al. performed comparative experiments on lateral heat diffusion in porcine muscles using monopolar and bipolar diathermy, the Harmonic Scalpel^TM^, and LigaSure^TM^. The application of monopolar diathermy (10 s at 40 W) resulted in a temperature of 59.2 °C in tissue 1 cm from the tip of the instrument. In contrast, both the Harmonic Scalpel^TM^ and the LigaSure^TM^ maintained temperatures below 20 °C at equivalent power levels [31]. The thermal spread of electrocautery has been reported to cause thermal damage to surrounding soft tissues at temperatures greater than 43 °C [32], meaning that even tissues 1 cm away can suffer significant thermal injury. Additionally, because the electrical conductivity of adipose tissue is approximately three times that of muscle, surrounding tissues are likely to reach higher temperatures during axillary dissection, which primarily involves adipose tissue [33]. Conventional electrocautery ALND often causes thermal damage to the surrounding lymphatic vessels, which should ideally be spared. This damage can lead to inflammation, eventual obstruction of lymphatic vessels, and ultimately the development of upper extremity lymphedema. In contrast, the TST with LGSED allows for ALND to be performed without causing excessive thermal damage to the surrounding tissue. Furthermore, by effectively sealing the lymphatic vessels [8], lymphatic leakage is minimized, and the drainage volume is reduced, which likely contributes to the prevention of BCRL. To the best of our knowledge, this is the first study to suggest that the monopolar electrocautery scalpel may be an important risk factor for the development of BCRL in breast cancer surgery with ALND. However, this technique has certain limitations. One notable drawback is that slight variations in surgical technique among operators may influence the outcomes. In particular, careful attention must be paid to the extent of lymph node dissection, such as preserving the lymphatic vessels around the axillary vein. In addition, the LigaSure™ Exact is a disposable device with a cost of approximately USD 670, making it relatively expensive. However, considering that it shortens the average hospital stay by 3.7 days compared to the conventional method and significantly reduces postoperative complications such as bleeding and seroma [8], it ultimately results in significant healthcare cost savings.

This study has notable limitations, including its single-center design, small sample size, and lack of randomization. However, these biases were mitigated by selecting consecutive clinical cases. Future research should aim to validate whether the TST with LGSED truly reduces the risk of BCRL through multicenter randomized controlled trials (RCTs). In such studies, it is essential to standardize the extent of lymph node dissection and ensure adequate training for the correct use of LGSED.

## 5. Conclusions

This preliminary study suggests that ALND with a bipolar-vessel-sealing system may significantly reduce the incidence of BCRL. However, further multi-institutional randomized controlled trials are necessary to confirm the reproducibility of these findings. If validated, this technique has the potential to become an essential surgical procedure for axillary dissection in breast cancer treatment in the near future.

## Figures and Tables

**Table 1 cancers-17-01285-t001:** Clinicopathologic characteristics of the study population.

	Total Sealing Technique (TST)	Conventional Method (CONV)	*p*
Number of patients	35	36	
Age	60.6 ± 14.1	66.2 ± 13.0	0.114
BMI (kg/m^2^)	24.4 ± 4.4	23.2 ± 4.1	0.337
Radiotherapy	14 (40.0%)	15 (41.7%)	>0.999
Neoadjuvant chemotherapy	18 (51.4%)	11 (30.6%)	0.074
Postoperative chemotherapy	10 (28.6%)	14 (38.9%)	0.454
Taxane containing regimen	26 (74.2%)	21(58.3%)	0.211
Total number of removed lymph nodes	17.3 (14.5–20.1)	15.3 (13.7–16.9)	0.119
Stage			**0.015**
0	0	2	
IA	1	4	
IIA	10	10	
IIB	5	9	
IIIA	6	7	
IIIB	3	2	
IIIC	5	1	
IV	5	1	

*p*-value < 0.05 was significant (in bold).

**Table 2 cancers-17-01285-t002:** (**A**) Comparison of lymphedema incidence between TST group and CONV group. (**B**) Logistic regression analysis of lymphedema incidence.

(**A**)		
	**Total Sealing Technique (TST)**	**Conventional Method (CONV)**
Total Number of patients	35	36
Lymphedema		
(+)	1	8
(−)	34	28
Lymphedema incidence rate (%)	2.9	22.2
(**B**)		
**Variable**	**Odds Ratio (95% CI)**	** *p* **
Surgical Instrument (TST vs. Conventional)	0.103 (0.012–0.873)	0.037

Abbreviations: CI, Confidence Interval.

**Table 3 cancers-17-01285-t003:** The correlation between BCRL incidence and risk factors in the TST group.

	TST Group		
	BCRL (+)	BCRL (−)	*p*
	(*n* = 1)	(*n* = 34)	
Age, years			>0.999
<63	0	17	
≥63	1	17	
BMI category (kg/m^2^)			>0.999
<25	1	22	
≥25	0	12	
Radiotherapy received			>0.999
Yes	0	14	
No	1	20	
Neoadjuvant or Postoperative chemotherapy received			
Yes	1	27	>0.999
No	0	7	
Taxane containing regimen			>0.999
Yes	1	25	
No	0	9	
Total lymph nodes removed			>0.999
<19	0	17	
≥19	1	17	
Postoperative drainage volume, mL			>0.999
<270	0	17	
≥270	1	17	
Duration of drainage, days			>0.999
<5	0	17	
≥5	1	17	
Seroma			0.286
Yes	1	9	
No	0	25	

Fisher’s exact test indicated no significant difference between the two groups (*p* > 0.999), reflecting complete independence in the variable distribution.

**Table 4 cancers-17-01285-t004:** The correlation between BCRL incidence and risk factors in the conventional group.

	CONV Group		
	BCRL (+)	BCRL (−)	*p*
	(*n* = 8)	(*n* = 28)	
Age, years			**0.041**
<71	7	11	
≥71	1	17	
BMI category (kg/m^2^)			>0.999
<25	5	19	
≥25	3	9	
Radiotherapy received			0.236
Yes	5	10	
No	3	18	
Neoadjuvant or Postoperative chemotherapy received			**0.033**
Yes	8	16	
No	0	12	
Taxane containing regimen			**0.011**
Yes	8	13	
No	0	15	
Total lymph nodes removed			0.709
<15	3	13	
≥15	5	15	
Postoperative drainage volume, ml			**0.003**
<600	0	18	
≥600	8	10	
Duration of drainage, days			0.422
<7	2	12	
≥7	6	14	
Seroma			>0.999
Yes	6	19	
No	2	9	

*p*-value < 0.05 was significant (in bold).

## Data Availability

The data presented in this study are available upon request from the corresponding author. The data are not publicly available because of ethical approval.

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
