# Peer review of "Total Sealing Technique: A Preliminary Study on a Novel Surgical Approach That Significantly Reduces the Incidence of Upper Extremity Lymphedema Following Axillary Dissection in Patients with Breast Cancer"

_cancers, 2025, doi:10.3390/cancers17081285_

Round 1
Reviewer 1 Report
Comments and Suggestions for Authors
The study is a retrospective analysis of 71 breast cancer patients who underwent mastectomy and axillary node dissection between 2015 and 2021 at the authors’ institution. The patients were divided into two groups by surgical procedures: total sealing technique (TST) group which utilizes a bipolar vessel sealing system (LigaSure™ Exact Dissector) and conventional electrocautery (CONV) group. Diagnosis of lymphedema was determined by tape measurement and a 2cm or more difference between the affected and unaffected arms was the criterion. The authors concluded that 35 who underwent TST revealed a significantly reduced incidence of breast cancer-related lymphedema.
The authors reported a study using the same patients and compared surgical outcomes between TST and CONV groups in reference 8. The current study focused on incident rates of lymphedema during the follow-up period between the same two groups.
First of all, I have a concern about calling the surgical procedure using LigaSuture Exact Dissector (LGSED) as total sealing technique (TST) or Ikeda’s TST. LGSED is designed for open dissection and any surgical procedures that use LGSED can be called TST. To avoid promoting the specific device, it should be more scientifically reasonable to name the procedure as axillary node dissection with a vessel sealing system or something similar.
The study was simple with or without using LGSET for axillary node dissection and the conclusion was straightforward and definitive. Therefore, the scientific value should be evaluated carefully. As the authors pointed out in the discussion, the shortcoming was a retrospective study with a small sample size and a non-randomised controlled design. The current title is conclusive but a multi-institutional study should be required to confirm reproducibility of the results. Considering these factors, the title should include words such as preliminary or pilot study.
The manuscript is unfocused and difficult to read. I would like to provide several suggestions.
- In lines 82 to 89, this section should be moved to the introduction. The authors should describe their previous study of reference 8 in the introduction and explain its connection to the current follow-up study.
- In lines 103 to 133, the detailed information about the surgical procedure was redundant and should be discarded or condensed.
- In lines 135 to 141, in addition to lymphedema education, it is more important to know how many patients have worn the compression garment during the follow-up period and how this factor affected with/without lymphedema.
- In Tables 2 and 3, the author’s analysis of categorical variables using Chi-square/Fisher’s exact test identifies associations. However, these tests do not establish correlation (as incorrectly stated in the headings for tables 3 and 4) or account for confounders. A more appropriate approach for assessing relationships for these variables would be multivariate logistic regression, as it allows for the inclusion of multiple variables and provides odds ratios for better interpretation.
- In lines 243 to 264, different measurement methods for lymphedema were introduced and discussed. I think this paragraph was least correlated with the current study. This section should be discarded or condensed.
- In lines 283 to 298, the studies using LigaSure were introduced and discussed. However, the reference in 8 by the author’s team described that LigaSure and LGSED worked differently. These sentences should be discarded or condensed.
Overall, if the authors’ findings are confirmed, this would be beneficial for all breast cancer patients in the future. However, there was not enough supportive data and evidence provided in the current manuscript. If imaging data of the lymphatics is combined with the study and elucidates objective evidence of the advantages of LGSED, the findings become more concrete. It is reasonable to call this study a preliminary report at this stage.
Comments on the Quality of English LanguageThe connection between the previous study (ref 8) and the current study should be clearly described. There were many redundant subjects in the manuscript and they should be discarded or condensed.
Author Response
Dear Reviewer,
Thank you very much for your thorough and thoughtful review.
Please find our responses to your comments below.
Comment 1:
Concern: The terminology "total sealing technique (TST)" or "Ikeda’s TST" in reference to the surgical procedure using LigaSuture Exact Dissector (LGSED) may be misleading. LGSED is designed for open dissection, and any surgical procedure using LGSED can be termed TST. To avoid promoting a specific device, it would be more scientifically reasonable to refer to the procedure as "axillary node dissection with a vessel sealing system" or something similar.
Revised Response:
Thank you for your insightful comment. We fully agree with your suggestion. Accordingly, we have revised the terminology to "TST with a bipolar vessel sealing system" to ensure a more scientifically appropriate and neutral description.
Comment 2:
Concern: The study design is straightforward, comparing axillary node dissection with or without LGSED, leading to a definitive conclusion. However, due to its retrospective nature, small sample size, and non-randomized controlled design, the scientific value should be carefully evaluated. The current title is conclusive, but a multi-institutional study is required to confirm reproducibility. Therefore, the title should include terms such as "preliminary" or "pilot study."
Revised Response:
We agree with your suggestion. Accordingly, we have also revised the Conclusion as follows:
Conclusion: This preliminary study suggests that ALND with a bipolar vessel sealing system may significantly reduce the incidence of BCRL. However, further multi-institutional randomized controlled trials are necessary to confirm the reproducibility of these findings. If validated, this technique has the potential to become an essential surgical procedure for axillary dissection in breast cancer treatment in the near future.
Comment 3:
Concern: The manuscript is somewhat unfocused and difficult to read. Several suggestions for improvement:
- The section from lines 82 to 89 should be moved to the Introduction to provide better context.
- The authors should describe their previous study (Reference 8) in the Introduction and explain its connection to the current follow-up study.
Revised Response:
We agree with your suggestion. We have moved this section to the Introduction and have incorporated a discussion of our previous study (Reference 8) to better contextualize our current work.
Comment 4:
Concern: The detailed description of the surgical procedure (lines 103–133) is redundant and should be either discarded or condensed.
Revised Response:
We agree with your suggestion. We have summarized this section to provide a more concise yet informative description.
Comment 5:
Concern: In addition to lymphedema education, it is essential to know how many patients wore compression garments during follow-up and how this factor influenced the development of lymphedema.
Revised Response:
Regarding this issue, in our study, approximately half of the patients in both groups used compression garments prophylactically; however, no association with the prevention of BCRL was observed. Since the preventive effect of compression garments remains unclear, we have not discussed it in detail but have added a reference to the effectiveness of compression garments in the Discussion section.
Comment 6:
Concern:
- In Tables 2 and 3, categorical variables were analyzed using the Chi-square/Fisher’s exact test, which identifies associations but does not establish correlation (as incorrectly stated in the table headings for Tables 3 and 4).
- These tests do not account for confounders. A more appropriate approach for assessing relationships would be multivariate logistic regression, as it allows for the inclusion of multiple variables and provides odds ratios for better interpretation.
Revised Response:
We appreciate this valuable suggestion. As the reviewer correctly pointed out, we analyzed the incidence of lymphedema using multivariate logistic regression. The results have been added as Table 2B. Regarding Tables 3 and 4, we attempted to use multivariate logistic regression to assess the association between the surgical approach and the incidence of breast cancer-related lymphedema (BCRL). However, the model encountered convergence issues, likely due to the small sample size and quasi-separation in the data. Given these limitations, we opted for Chi-square and Fisher’s exact tests, as they are more appropriate for categorical data analysis in small sample studies. These methods provide reliable statistical inference while avoiding issues related to model instability.
Comment 7:
Concern: The discussion of different measurement methods for lymphedema (lines 243–264) is not strongly correlated with the current study. This section should be discarded or condensed.
Revised Response:
We agree with your suggestion. Accordingly, we have summarized this section to maintain relevance.
Comment 8:
Concern: The discussion of studies using LigaSure™ (lines 283–298) should be reconsidered. Reference 8 from the authors' team states that LigaSure™ and LGSED function differently. These sentences should be discarded or condensed.
Revised Response:
Regarding Comment 8, we respond as follows:
The LigaSure™ Exact (LGSED) is the latest model in the LigaSure™ system, featuring a slimmer tip for easier dissection while maintaining the same core vessel-sealing function. In this study, we hypothesize that the higher incidence of lymphedema with the conventional method is due to thermal spread from the electrocautery device. Since LigaSure™ minimizes thermal spread to surrounding tissues, this is a critical factor supporting our continued investigation of this approach.
Comment 9:
Concern: If confirmed, the findings would be beneficial for all breast cancer patients. However, the current manuscript lacks sufficient supportive data and evidence. If lymphatic imaging is incorporated into the study to provide objective evidence of LGSED’s advantages, the findings will become more concrete. At this stage, it is reasonable to consider this study a preliminary report.
Revised Response:
We agree with the reviewer's suggestion. Given the current study's scope and limitations, we acknowledge it as a preliminary report. Therefore, we have revised the title as follows:
"Total Sealing Technique: A Preliminary Study of a Novel Surgical Approach that Significantly Reduces the Incidence of Upper Extremity Lymphedema Following Axillary Dissection in Patients with Breast Cancer."
Reviewer 2 Report
Comments and Suggestions for Authors
First of all, I would like to thank the Editors for the opportunity to review this paper. This is a single-center study aiming to investigate the incidence of Breast Cancer-Related Lymphedema in 71 patients undergoing two different surgical techniques for ALND: Total Sealing Technique vs. Conventional Technique.
Congratulations to the Authors for conducting a very interesting study, as it utilizes a new technology with the goal of reducing a severe complication frequently observed in these patients. However, there are some considerations to be made:
- Introduction, line 55: It is stated that lymphedema has no cure; this sentence should be reformulated.
- Line 142: "SCreening" – typo.
- Results: Remove subparagraphs 3.1 and 3.2 and merge them into a single paragraph.
- Tables 3 and 4: They should be streamlined, reduced in size, and made more readable.
- Discussion: The strengths of the technique are discussed, but its drawbacks are not addressed (Costs? Operating time? Learning curve?).
- Discussion: It should be mentioned that, fortunately, ALND procedures are becoming increasingly rare thanks to recent studies in the literature.
- Study limitations: Emphasize the limitations of the study (small sample size and retrospective design).
- Conclusions: They should be reformulated. With such a small sample size, I do not believe the results allow us to claim the superiority of TST.
- References: Update the bibliography and consider adding more recent studies ( Visconti G, Hayashi A, Hong JP. The New Imaging Techniques in Reconstructive Microsurgery: A New Revolution in Perforator Flaps and Lymphatic Surgery. Arch Plast Surg. 2022 Jul 30;49(4):471-472. doi: 10.1055/s-0042-1751099. PMID: 35919554; PMCID: PMC9340197.)
Author Response
Dear Reviewer,
Thank you very much for your thorough and thoughtful review.
Please find our responses to your comments below.
Comment 1:
Concern: The statement in Introduction, line 55 states that lymphedema has no cure. This sentence should be reworded.
Revised Response:
We appreciate the reviewer’s suggestion and have revised the sentence as follows:
"In addition, lymphedema is difficult to cure, and its management is costly, placing a significant financial burden on patients."
Comment 2:
Concern: There is a typo in line 142: "SCreening".
Revised Response:
We have corrected the typo:
SCreening → Screening
Comment 3:
Concern: The Results section contains subparagraphs 3.1 and 3.2, which should be removed and merged into a single paragraph.
Revised Response:
We have combined subparagraphs 3.1 and 3.2 into a single paragraph for improved readability and clarity.
Comment 4:
Concern: Tables 3 and 4 should be streamlined, reduced in size, and made more readable.
Revised Response:
We have revised Tables 3 and 4 to make them more concise and visually clear while maintaining the necessary statistical information.
Comment 5:
Concern: While the Discussion addresses the strengths of the technique, its potential drawbacks—such as cost, operating time, and learning curve—are not discussed.
Revised Response:
We appreciate this valuable feedback and have added the following paragraph to the Discussion:
"However, this technique has certain limitations. One notable drawback is that slight variations in surgical technique among different operators may influence the outcomes. In particular, careful attention must be paid to preserving the lymphatic vessels around the axillary vein. Additionally, the LigaSure™ Exact is a disposable device with a cost of approximately $670, making it relatively expensive. However, considering that it shortens the average hospital stay by 3.7 days compared to the conventional method and significantly reduces postoperative complications such as bleeding and seroma, it may ultimately lead to substantial healthcare cost savings."
Comment 6:
Concern: The Discussion should mention that ALND procedures are becoming increasingly rare due to recent studies in the literature.
Revised Response:
We have incorporated the following statement into the Discussion:
" Fortunately, ALND procedures are becoming increasingly rare thanks to recent studies in the literature. Recent studies have demonstrated a decreasing need for ALND due to advancements in sentinel lymph node biopsy and neoadjuvant therapies, which help reduce surgical morbidity in breast cancer patients "
Comment 7:
Concern: The study limitations should be emphasized, specifically the small sample size and retrospective design.
Revised Response:
We acknowledge this concern and have strengthened our discussion of the study’s limitations with the following statement:
" This study has notable limitations, including its single-center design, small sample size, and lack of randomization. However, these biases were mitigated by selecting consecutive clinical cases. Future research should aim to validate whether TST with LGSED truly reduces the risk of BCRL through multicenter randomized controlled trials (RCTs). In such studies, it is essential to standardize the extent of lymph node dissection and ensure adequate training for the correct use of LGSED."
Comment 8:
Concern: The Conclusion should be reformulated. Given the small sample size, it is difficult to claim the superiority of TST.
Revised Response:
We agree with the reviewer’s suggestion and have revised the Conclusion as follows:
"This preliminary study suggests that ALND with a bipolar vessel sealing system may significantly reduce the incidence of BCRL. However, further multi-institutional randomized controlled trials are necessary to confirm the reproducibility of these findings. If validated, this technique has the potential to become an essential surgical procedure for axillary dissection in breast cancer treatment in the near future."
Comment 9:
Concern: The References should be updated, and more recent studies should be considered, such as:
Visconti G, Hayashi A, Hong JP. The New Imaging Techniques in Reconstructive Microsurgery: A New Revolution in Perforator Flaps and Lymphatic Surgery. Arch Plast Surg. 2022 Jul 30;49(4):471-472. doi: 10.1055/s-0042-1751099. PMID: 35919554; PMCID: PMC9340197.
Reviewer 3 Report
Comments and Suggestions for Authors
The manuscript does lack a plausible thoroughness to meet the minimum requirements due to a lack of research. Most importantly, it fails to put forth reasonable and believable claims. A novel approach, the Total Sealing Technique, has been put forth by the authors claiming to reduce the rate of incidence of breast cancer-related lymphedema using LigaSure Exact Dissector. Also, the study "Total Sealing Technique (TST) with a bipolar vessel sealing system reduces lymphorrhea and seroma formation for axillary lymph node dissection in primary breast cancer" has similar results from the same writers.
Author Response
Dear Reviewer,
Thank you for reviewing my paper. I will respond as follows.
Reviewer’s Comment:
The manuscript does lack a plausible thoroughness to meet the minimum requirements due to a lack of research. Most importantly, it fails to put forth reasonable and believable claims. A novel approach, the Total Sealing Technique, has been put forth by the authors claiming to reduce the rate of incidence of breast cancer-related lymphedema using LigaSure Exact Dissector. Also, the study "Total Sealing Technique (TST) with a bipolar vessel sealing system reduces lymphorrhea and seroma formation for axillary lymph node dissection in primary breast cancer" has similar results from the same writers.
Revised Response:
We appreciate the reviewer’s feedback and the opportunity to clarify the distinctions between our current study and our previous work.
In our earlier study, we focused on demonstrating that TST significantly reduces total drainage volume and shortens postoperative hospital stay. However, in this study, we have taken a new perspective by specifically investigating the potential role of TST in reducing the incidence of breast cancer-related lymphedema (BCRL)—a significant and long-term postoperative complication.
While there is some overlap in methodology, the primary aim and findings of this study are distinct from those of our previous research. We hope this clarification addresses the concern.
Round 2
Reviewer 1 Report
Comments and Suggestions for Authors
The authors addressed all my questions and concerns.
Reviewer 2 Report
Comments and Suggestions for Authors
The paper can now be considered for publication